# Plasma A_2A_R Measurement Can Help Physicians Identify Patients Suspected of Coronary Chronic Syndrome: A Pilot Study

**DOI:** 10.3390/biomedicines10081849

**Published:** 2022-08-01

**Authors:** Franck Paganelli, Gabriel Cappiello, Soumeya Aliouane, Nathalie Kipson, Christine Criado, Khadidja Hamou, Jehuel Ntawanga, Erika Peroni, Maria Carreno, Lucas Methlin, Giovanna Mottola, Julien Fromonot, Pierre Deharo, Marine Gaudry, Marion Marlinge, Régis Guieu, Jean Ruf

**Affiliations:** 1Department of Cardiology, North Hospital, 13015 Marseille, France; franck.paganelli@ap-hm.fr (F.P.); gabriel.cappiello@ap-hm.fr (G.C.); soumeya.aliouane@ap-hm.fr (S.A.); khadidja.hamou@ap-hm.fr (K.H.); jehuel.ntawanga@ap-hm.fr (J.N.); 2Center for Cardio-Vascular and Nutrition Research (C2VN), INSERM, INRAE and Aix-Marseille University, 13005 Marseille, France; nathalie.kipson@univ-amu.fr (N.K.); christine.criado@univ-amu.fr (C.C.); giovanna.mottola@ap-hm.fr (G.M.); julien.fromonot@univ-amu.fr (J.F.); pierre.deharo@ap-hm.fr (P.D.); marion.marlinge@ap-hm.fr (M.M.); jean.ruf@univ-amu.fr (J.R.); 3Laboratory of Biochemistry, Timone University Hospital, 13005 Marseille, France; erika.peroni@ap-hm.fr (E.P.); maria.carreno@ap-hm.fr (M.C.); lucas.methlin@ap-hm.fr (L.M.); 4Department of Cardiology, Timone University Hospital, 13005 Marseille, France; 5Department of Vascular Surgery, Timone University Hospital, 13005 Marseille, France; marine.gaudry@ap-hm.fr

**Keywords:** soluble A_2A_ adenosine receptors, coronary artery disease, ELISA

## Abstract

The evaluation of suspected coronary artery disease (CAD) in the medical community is challenging. Patients with suspected coronary chronic syndrome (CCS) are referred by the medical community to be assessed by specialists for the performance of noninvasive tests that have high rates of false positives and false negatives. While troponins are the gold standard for evaluate myocardial injuries, there is no biomarker to assess myocardial ischemia in patient populations with negative electrocardiography or without an increase in troponin level. A_2A_ adenosine receptors control the coronary blood flow through its vasodilating properties. It has been shown that patients with CAD have a lower A_2A_R expression on peripheral blood mononuclear cells, suggesting a link between A_2A_R production and the severity of CAD. Herein, we present a new and innovative method of inhibition ELISA for A_2A_R in the plasma of patients who permit the evaluation of the amount of soluble A_2A_R. For this analysis, the total study sample was 54, including 31 patients with CAD with stenosis > 50% and a significant fractional flow reserve (FFR < 0.8) (Group 1) and 23 patients with normal or non-obstructive coronary arteries (stenosis < 50% and nonsignificant FFR > 0.8) (Group 2). The % inhibition (which is linked to the presence of soluble receptors) with the plasma of patients with FFR < 0.8 was significantly lower than that of patients with FFR > 0.8 (median [range]: 68% [20.7–86.9] vs. 83% [67–88.4]; *p* < 0.001). The ROC curve indicated a good sensitivity/specificity ratio with a cut off of 72.5% and an area under the curve of 0.87. In conclusion, a rapid ELISA to assess soluble A_2A_R in the plasma shows promise to screen patients suspected of having CAD.

## 1. Introduction

Chronic chest pain is a common presenting symptom in primary care, of which there are many possible causes. The differential diagnosis of chronic chest pain is wide-ranging and includes cardiac pain as exertional chest pain or chronic coronary syndrome (CCS). The diagnosis of coronary heart disease (CHD) may be missed in these patients [1]. ESC guidelines recommend that stable angina can be assessed by a specialist within 2 weeks of referral [2]. However, the evaluation of patients with stable symptoms and suspected CHD (stable angina) may be a challenge. The European Society of Cardiology (ESC) recommends the use of clinical risk scores to calculate pretest probability, as determined by the Diamond-Forrester Coronary Artery Surgery Study [2] and the updated version of the Coronary Artery Disease Consortium (CADC) clinical risk scores, followed by the use of functional testing [2]. All of these noninvasive tests have high rates of false positives and false negatives [2,3]. At certain times, the ability to recognize high-risk patients can be difficult through the sole use of clinical guidelines, and a symptom-based clinical assessment followed by coronary computed tomography angiography (CCTA) is needed to identify obstructive and nonobstructive CAD, demonstrating more benefits over the use of standard testing [4,5,6]. Biomarkers, such as high-sensitivity troponin, heart-type fatty acid binding protein, copeptin, or ischemia-modified albumin, constitute an important diagnostic advancement among patients with chest pain. Due to the fact that some biomarkers lack specificity (copeptin and ischemia-modified albumin) and specific biomarkers (such as high-sensitivity troponin) are markers of acute myocardial injury, biomarker assays are not currently used in the investigation of patients with suspected stable CCS [2].

Adenosine, through the activation of its A_2A_ and A_2B_ receptors (A_2A_R and A_2B_R), controls coronary blood flow [7]. Vasodilatation occurs via (i) cAMP production in smooth muscle cells, cAMP production, and coronary artery dilation [8]; (ii) NO release [9]; and (iii) the activation of K_ATP_ channels [10,11]. Recently, it has been shown that the decrease in A_2A_R production on peripheral blood mononuclear cells (PBMCs) correlates with the severity of CAD [12,13]. Thus, the measurement of A_2A_R production on PBMCs can help in screening CAD patients on a large scale. However, the test conditions require a long procedure that is not compatible with rapid evaluation and which makes it difficult to routinely use on a large scale. Therefore, the aim of this study was to assess the measurement of soluble A_2A_R in the plasma for the investigation of CCS in patients with suspected stable angina who were referred for invasive coronary angiography (ICA) in the cardiology department of the North Hospital in Marseille, France. Due to the fact that ischemia impacts A_2A_R expression on PBMCs and that the decrease in this expression correlates with the severity of CAD [12,13], we hypothesize that such a decrease could be found directly in the plasma for the expression of soluble A_2A_R.

## 2. Materials and Methods

### 2.1. Patient Selection

We conducted a blind prospective study on patients with suggestive CCS based on a clinical assessment, including intermediate PTP (15–85%) with normal echocardiography (left ventricular ejection fraction [LVEF] >50%) and negative electrocardiography and troponin at admission, as well as negative ECG-troponin monitoring. We excluded patients with (i) a ST segment elevation of ≥1 mm in 2 contiguous leads on the presenting ECG; (ii) any significant increase in cardiac troponin (>99th percentile); (iii) pretest probability < 15%; (iv) Global Registry of Acute Coronary Events (GRACE) score of >140, known CAD, or awaiting revascularization (v) history and clinical examinations suggesting noncardiac chest pain; (vi) abnormal resting echocardiography and/or chest X-ray, as well as hemoglobin < 13 g/dL; and (vii) thyroid disorder. No sex-based or race/ethnicity-based differences were addressed. Smoking history status was self-reported on the based on a questionnaire that was submitted at the time of hospitalization. The study was approved by the ethics committee. When the patient was eligible, the physician scheduled at least one stress imaging test or abnormal coronary computed tomography (CCTA). When inducible myocardial ischemia in at least one of the stress imaging tests or abnormal CCTA was observed, invasive strategies, such as invasive coronary angiography (ICA) and flow fraction reserve (FFR) (if necessary) were intended. After informed consent (for the ICA, FFR and blood sample assays) was obtained, patients were included in the study (see Figure 1). Any patient wishing to withdraw consent during the study was excluded. The timing of ICA was determined by the physician and carried out according to the guidelines; however, this was performed after the performance of at least one stress imaging test. Venous blood samples for the A_2A_R assay were obtained just before ICA and FFR. The biologist was not aware of the results of the invasive and noninvasive tests. Similarly, this was also carried out for the cardiologist regarding the results of the A_2A_R assays. An independent Clinical Endpoints Committee (CEC) and core labs who were blinded to the results adjudicated CAD with quantitative coronary angiography (QCA) results and FFR results for obstructive CAD.

### 2.2. ICA and FFR

ICA was performed according to a standard clinical method by using a visual quantitative scoring system for the image analysis, with CAD defined as a luminal diameter narrowing between 20% and 90% in one or more epicardial arteries or their major branches. Vessels with a luminal diameter <2 mm were excluded. Intracoronary glyceryl trinitrate (200 µg) was injected to minimize the risks of vasospasm. When arteries with stenosis >20% were visually perceived, an FFR pressure wire (Certus, St. Jude Medical, St. Paul, MN, USA) was positioned distal to the stenosis of interest to determine vessel FFR by using a Radio-Analyzer (St. Jude Medical, St. Paul, MN, USA) under steady-state hyperemia (intravenous adenosine: 140 µg/kg/min for 3–6 min). A FFR ≤ 0.80 was considered an evidence-based physiological threshold indicative of obstructive CAD in clinical practice to perform percutaneous coronary intervention [14,15].

### 2.3. Adjudicated Final Diagnosis

To establish the final diagnosis at discharge for each patient, two independent cardiologists who were blinded to the results of A_2A_R pharmacological characteristics reviewed all of the available medical records (including patient history, physical examination, the results of laboratory tests, exercise stress testing, ICA, and FFR) from the time of cardiac department presentation to discharge. In cases of diagnosis disagreement, data were reviewed and adjudicated in conjunction with a third cardiologist.

### 2.4. Soluble Plasma A_2A_R Measurement

#### 2.4.1. Sample Collection

Blood was collected via venipuncture at the brachial vein in heparinized tubes. Blood samples without visible hemolysis were centrifuged at 3000× *g* for 10 min at room temperature to remove the cells. The supernatant plasma was transferred to Eppendorf tubes and centrifuged again at 10,000× *g* for 30 min at room temperature to remove any residual protein aggregates, platelets, cell debris, and large vesicles. Plasma samples were immediately used or frozen at −80 °C.

#### 2.4.2. A_2A_R Peptide

A 30 residue-long peptide (NNCGQPKEGKNHSQGCGEGQVACLFEDVVP; residues 144–173) corresponding to the second extracellular loop of human A_2A_R (UniProtKB/Swiss-Prot Entry P29274) was synthesized at >95% purity by Genosphere-Biotechnologies (Paris, France). Reverse-phase HPLC gives one eluted peptide peak at 220 nm, which resolves by mass spectrometry (MALDI-TOF) at a molecular weight of 3146.

#### 2.4.3. Inhibition ELISA

The A_2A_R identification and quantification experiments were performed by using inhibition ELISA. Each plasma sample was incubated with Adonis, which is a specific homemade monoclonal antibody (A_2A_R mAb) that binds to a linear epitope from the second extracellular loop of human A_2A_R with high affinity [16]. After incubation, the antibody-antigen mixture was added to the plate coated with the A_2A_R peptide (0.1 μg/well, incubated overnight at 4 °C under a humidified atmosphere) to allow for the free A_2A_R mAb to bind with it. As a result of the prior binding of A_2A_R from the plasma sample to the A_2A_R mAb, the reaction in the ELISA plate wells is reduced, and the reduction in absorbance (A) in the wells is inversely proportional to the concentration of A_2A_R in the test sample. In our study, the following protocol was used to develop the inhibition ELISA. The plasma dilution and A_2A_R mAb concentration that were used in the assay were optimized by using a checkerboard analysis to ascertain a suitable assay working range, while keeping the antigen–antibody reaction in the zone of equivalence and minimizing the number of materials that were used in the assay. In this assay, a predetermined dose of Adonis was previously incubated in an Eppendorf tube for 90 min under shaking at room temperature (RT) with a 1/30 dilution of the plasma sample. Subsequently, each well in the A_2A_R peptide coated-microtiter plate received 100 μL of preincubated mixture corresponding to 0.13 μg of A_2A_R mAb and 3.3 μL of the plasma sample after the saturation step with bovine serum albumin. Assays were performed in quadruplicate, and wells filled with buffer alone served as blanks. After 90 min of incubation at RT, the plate was washed and incubated for 1 h at RT with 100 μL/well of appropriately diluted alkaline phosphatase-labeled anti-mouse antibodies. The plate was washed again and incubated for 20 min at RT with 200 μL/well of *p*-nitrophenylphosphate buffer substrate. In each well, A was measured at 405 nm on an ELISA reader. The “Reference sample” (A_ref_) consisted of coated wells incubated with A_2A_R mAb without any preincubation with the test sample and had the maximum A. The percentage inhibition of A_ref_ in the different wells containing a patient sample was calculated after the subtraction of the blank A value to all A values with the following formula: % I = (A_ref_−Patient sample A)/A_ref_ × 100.

### 2.5. Statistical Analysis

On the basis of the first values of % I calculated for patients FFR < 0.8 and FFR > 0.8 and the coefficient of variation of the ELISA, we considered in this pilot study, a reduction of at least 20% of the value of % I in Group 1 (FFR < 0.8) compared to Group 2 (FFR > 0.8) in order to estimate the minimum number of subjects to enroll (sample size = 20 for each Group is sufficient), which was calculated with a two-sided type-I error α equal to 0.05 and a power equal to 90%. Categorical variables are reported as numbers and percentages, and quantitative variables are reported as the means and standard deviations (SDs) or as medians and interquartile ranges (IQRs). A descriptive analysis was first performed according to the 2 groups of interest. For example, descriptive analyses were performed for Group 1 patients with a normal or subnormal angiogram with nonsignificant FFR and for Group 2 CAD patients with significant stenosis (>50%) and significant FFR < 0.8. The characteristics were compared between the 2 groups by using the chi-square test or Fisher’s test for categorical variables and the Kruskal–Wallis test for quantitative variables. The number of disease vessels was compared between groups by using the chi-square test, whereas the Mann–Whitney test was used to compare biological data between groups. The receiver operating characteristic (ROC) curve and area under the receiver operating characteristic curve (AUROC) values were established to define the best threshold value for soluble A_2A_R levels to discriminate patients between the 2 groups. The areas under the curve and their 95% confidence intervals were estimated. Sensitivity and specificity associated with the best threshold were estimated according to the Youden method [17], which allows for the maximization of both values. An AUROC from 0.9 to 1 was considered to indicate excellent accuracy, 0.8 to 0.9 was considered to be very good, 0.7 to 0.8 was considered to be good, 0.6 to 0.7 was considered to be sufficient, and <0.6 was considered to be insufficient. All tests were 2-sided, and *p* < 0.05 was statistically significant. All tests were performed by using R software (Microsoft, Redmond, Washington, DC, USA, Version 3.4.1). Technicians performing the biological analysis and the medical staff participating in the study were blinded to the clinical and biological results, respectively.

## 3. Results

### 3.1. Study Population

Sixty-nine patients with suspected CCS with negative electrocardiography and troponin were consecutively enrolled in this prospective pilot study according to the hospital protocol. We focused on those subjects who had available information. The study sample of 54 patients was finally analyzed (1 September–29 October 2020) (see flowchart, Figure 2), including 31 CAD patients with stenosis >50% and FFR < 0.8 (FFR+) (Group 1) and 23 patients with strictly normal coronary arteries or non-obstructive coronary arteries (stenosis < 50% and FFR > 0.8 (FFR−)) (Group 2).

The demographic and angiography characteristics are detailed in Table 1. No significant differences were observed between the two groups concerning age, sex ratio, or body mass index.

We observed an increasing prevalence and number of CAD risk factors (such as diabetes mellitus and smoking status) and/or more treatment for T2DM (DPPIV inhibitors) and dyslipidemia (statins) in Group 1. All fifty-four patients were investigated via ICA, but five patients in Group 2 were not selected for FFR testing because their arteries did not have identifiable atheroma plaques. FFR testing was successfully performed in the remaining 18 patients in Group 2 with visually perceived diameter stenosis between 30% and 50%, which involved 30 subtended territories. In this case, all of the FFR statuses were negative (such as FFR > 0.8). No completely or partially occluded arteries were found in Group 1. FFR testing was successfully performed in all patients with visually perceived diameter stenosis >50%. Only 31 patients (Group 1) were considered to present with significant CAD. Two patients in Group 2 were considered to have non-hemodynamically significant coronary artery stenosis despite having visually perceived diameter stenosis >70% (FFR > 0.8). By using this approach, FFR interrogations of 50 lesion arteries in Group 1 were classified as being significant in eighteen patients with single-vessel disease, seven patients with double-vessel disease, and six patients with triple-vessel disease. We performed noninvasive diagnostic tests in each of these 54 patients without a significant difference between the two groups.

### 3.2. Inhibition ELISA

To establish the test conditions, serial dilutions of buffer with a saturating amount of A_2A_R mAb (Adonis) alone and mixed with one representative plasma from patients with (FFR < 0.8) or without (FFR > 0.8) stenosis were performed (Figure 3). Blank includes buffer without A_2A_R mAb, and plasma results are expressed in A readings at 405 nm and are the mean of the quadruplicates (CV < 10%). A dose–response curve of A_2A_R mAb decreased in the presence of plasma from patients. The curve with plasma from patients with normal coronary angiography (FFR < 0.8) was lower than the curve from patients with stenosis. This indicated a preponderant presence of A_2A_R in FFR > 0.8 patients and a decreased level of A_2A_R in FFR < 0.8 patients. Based on these curves, the more discriminating dilution between the two patients was 1/27. At this dilution, the calculated % I values (as reported in the Methods section) for FFR < 0.8 and FFR > 0.8 patients were 26.5% and 67.3%, respectively.

Patients from the two groups were tested by using inhibition ELISA for A_2A_R in the plasma. Each test was performed in quadruplicate and reproduced twice at an interval of one week. For the selected population of 54 CAD patients, the intra- and inter-assay coefficients of variation were consistently <10% and <20%, respectively. Expressed in % I of Adonis mAb binding to the A_2A_R peptide, it appeared that A_2A_R in the plasma from Group 1 (FFR < 0.8) patients was significantly less abundant than in plasma from Group 2 (FFR > 0.8) patients (median [range]: 68% [20.7–86.9] vs. 83% [67–88.4]; *p* < 0.001) (Figure 4).

ROC curves showed a good sensitivity/specificity ratio with a cutoff of 72.5% I for the novel ELISA, with an AUROC of 0.87 considered to be very good. The 100% specificity was obtained with a cutoff of 65.5% I (Figure 5).

## 4. Discussion

This pilot study compared patients with an obstructive (FFR < 0.8) CAD and patients with normal or subnormal angiograms and those patients identified as having a significantly low level of A_2A_R expression in plasma from FFR < 0.8 patients compared to FFR > 0.8 patients by using a simple ELISA. From a clinical point of view, these results contribute to the emergence of a potential diagnostic tool in CCS. To our knowledge and for the first time, the current study described the use of soluble A_2A_R in the investigation of CCS. Although a decrease in A_2A_R expression on PBMCs in CAD [12,13] and the presence of A_2A_R reserves in ischemia inducible CAD (severe CAD) have been reported in CAD patients [18,19] via the use of a rapid ELISA method, we reported a decrease in soluble A_2A_R in plasma in patients with inducible ischemia. Although ELISA is a conventional method, its application for detecting soluble A_2A_R in the general population and especially in CAD patients has never been tested (to the best of our knowledge). This method may be innovative for screening CAD with a high risk of ischemia by using a single noninvasive venous blood collection procedure.

Suspected CCS remains a common presenting complaint. These patients are at a naturally high risk of developing cardiovascular events compared with the general population. In recent clinical trials, the annual mortality from CCS ranged from 0.9% to 2.9% [20]. A considerable number of CCS patients present with obstructive CAD with higher chances for future myocardial infarcts. The challenge remains to identify those patients with obstructive CAD who need further functional testing and/or ICA and those patients for whom myocardial ischemia can be ruled out with high specificity. Several biomarkers have been proposed for this scenario. The best known biomarkers are heart-type fatty acid-binding protein (hFABP), copeptin, hs-CRP, natriuretic peptides, ischemia-modified albumin (IMA), and GDF-15. However, these biomarkers lack sensitivity and specificity [20]. In previous studies, ultrasensitive cTnI values < 0.5 ng/L have ruled out functionally relevant CAD in 10% of tested patients [21]. Currently, a single cTnI value is not indicative of a high enough sensitivity and specificity (and negative and positive predictive values) for stable CAD, but it likely constitutes a sensitive marker for cardiac damage or abnormality [22].

Adenosine, through the activation of A_2A_R and A_2B_R, strongly impacts coronary blood flow [7,11]. It has been shown that PBMCs are a good model for evaluating the adenosinergic profile of these receptors, since the behavior of adenosine receptors expressed at the membrane of PBMCs mirrors the behavior of the adenosine receptor of the cardiovascular system. Thus, it has been shown that the expression and function of A_2A_R on PBMCs correlates with their expression and function (i.e., cAMP production by agonists) in the myocardium [23], coronary arteries [13], and femoral arteries [18]. Moreover, it has been shown that CAD patients exhibit low A_2A_R on PBMCs [12,13,19]. The low expression of A_2A_R can be observed in coronary artery disease patients. Indeed, the activation of A_2A_R leads to the dilation of the coronary arteries, and their weak production could limit coronary dilation, particularly during tests of effort [19]. A specific pharmacological profile of A_2A_R (known as the reserve receptor) has also been shown to be associated with myocardial ischemia [12,19]. Recently, the presence of ubiquitinated A_2A_R in extracellular vesicles from CAD patients has been reported [24]. When considering the systemic nature of A_2A_R expression, it is possible that a soluble part of A_2A_R circulates as exosomes in the plasma and reflects the activity of cellular A_2A_R. Herein, we found a low level of soluble A_2A_R in patients with severe CAD, which perfectly matches the low level of A_2A_R on PBMCs and the artery tissues of these patients (as has already been reported).

Due to the fact that there is no actual biomarker that predicts coronary events, the use of the novel A_2A_R ELISA has the potential to greatly improve patient care. We hope to be able to identify patients with suspected CAD with a less severe form of the disease and a good prognosis, thereby avoiding unnecessary invasive and noninvasive tests and revascularization procedures. Although the measurement of the expression level of receptors on PBMCs via Western blot requires a considerable amount of time (thus making it difficult to use on a large scale), the automation of our ELISA method could allow for screening on a large scale of patients suspected of having CAD.

This test can also identify patients with suspected CAD by indicating high risk and severe CAD. With an aging population and a rising burden of risk factors, such as obesity and diabetes, the very high prevalence of CCS results in a very high economic burden. Our plasma (extracellular vesicle) A_2A_R biomarker may be of major clinical interest for both patients and the health system to determine which diagnostic strategy is the most efficient for preventing recurrent ischemia and for reducing the morbidity and mortality of the disease. If the accuracy and the best threshold of these new biomarkers for identifying CAD are confirmed, it will lead to an improvement in the care of patients with CCS.

### Limitations of the Study

The prevalence of obstructive CAD in our cohort was lower than anticipated, which reduced the power of our analysis. A confirmation by using larger studies is warranted before the A_2A_R-ELISA can be adopted into clinical practice. Finally, although our population solely consisted of patients suspected of CAD, we cannot exclude the possibility that another inflammatory pathology may interfere with our assays by acting on the adenosinergic system and, consequently, on the A_2A_R.

## 5. Conclusions

A rapid inhibition method based on ELISA for evaluating soluble A_2A_R in the plasma may be helpful for screening patients suspected of having CAD. The specificity of the new method is ensured by the coupling of the A_2A_R peptide and a mAb against it, and the reproducibility of ELISA is well suited for use in clinical routines.

## Figures and Tables

**Figure 1 biomedicines-10-01849-f001:**
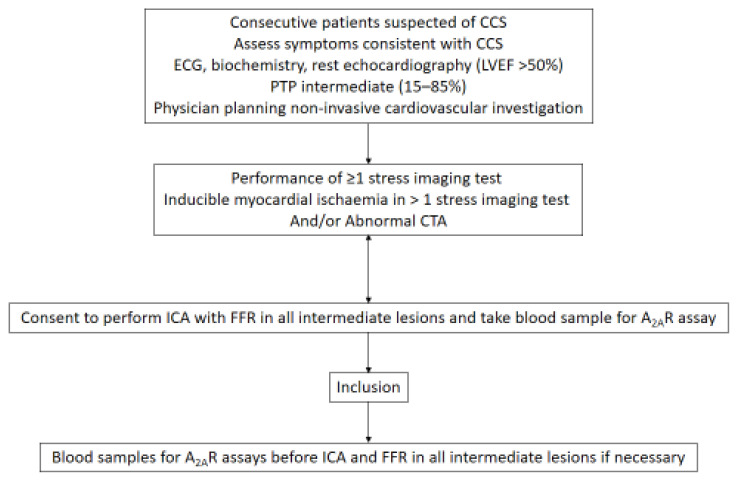
Flowchart methods. Patients with positive imaging stress, abnormal computed tomography angiography (CTA), or both were included. CCS: suspected coronary syndrome. PTP: pretest probability.

**Figure 2 biomedicines-10-01849-f002:**
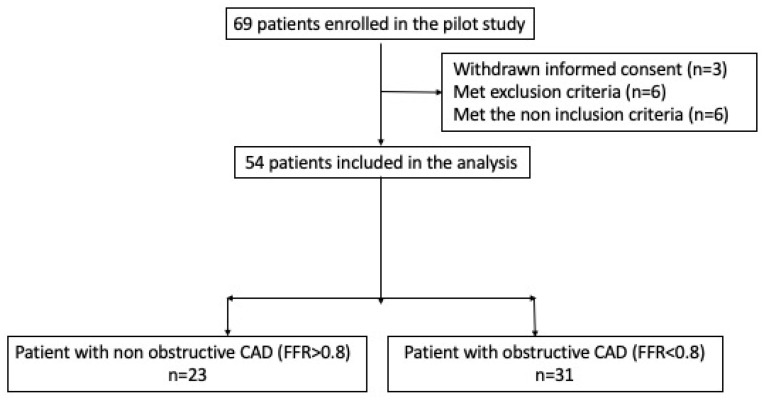
Flowchart results. Among the 69 patients who met the inclusion conditions, only 54 were finally able to be selected for the pilot study.

**Figure 3 biomedicines-10-01849-f003:**
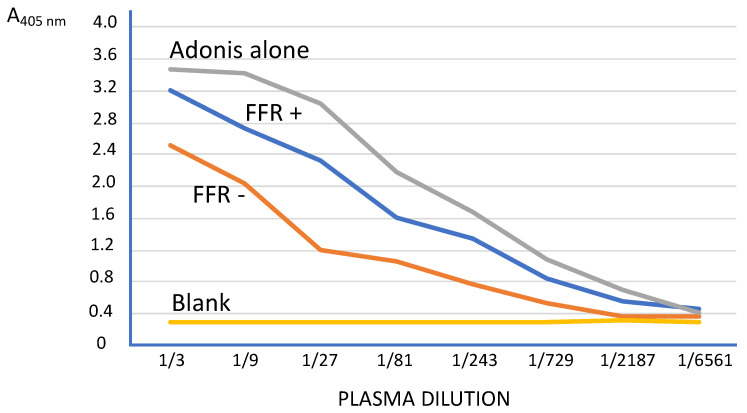
Inhibition ELISA for A_2A_R in the plasma of patients. Serial dilution curves obtained with a saturating amount of A_2A_R mAb (Adonis) alone and mixed with one representative plasma from patients with FFR < 0.8 (FFR+) or without stenosis FFR− (FFR > 0.8). The blank included buffer without A_2A_R mAb and plasma. The results are expressed in Absorbance (A) readings at 405 nm and are the mean of the quadruplicates (CV < 10%).

**Figure 4 biomedicines-10-01849-f004:**
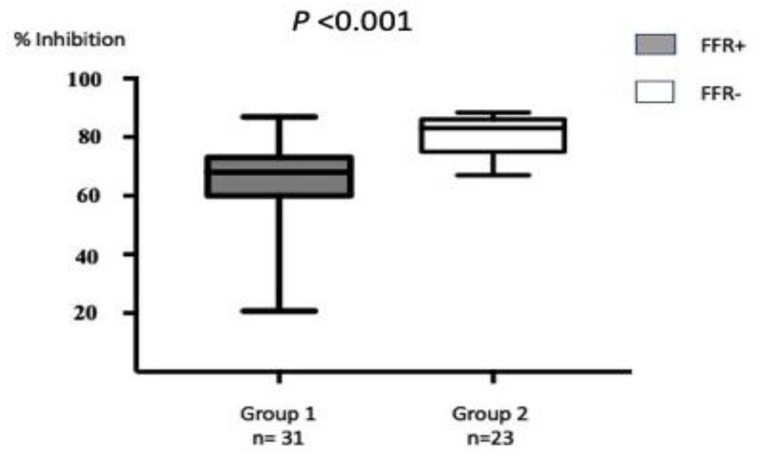
Comparative expression of soluble A_2A_R in the plasma from patients FFR+ and FFR−. The results from the inhibition ELISA are expressed in % I (see Methods). In this test, the more A_2A_R that was present in the sample indicated a higher % I. FFR < 0.8 (FFR+) patients had a lower % I than FFR− (FFR > 0.8) patients, which indicated that they expressed less A_2A_R than the other patients. The results obtained in the two groups were significantly different (*p* < 0.001).

**Figure 5 biomedicines-10-01849-f005:**
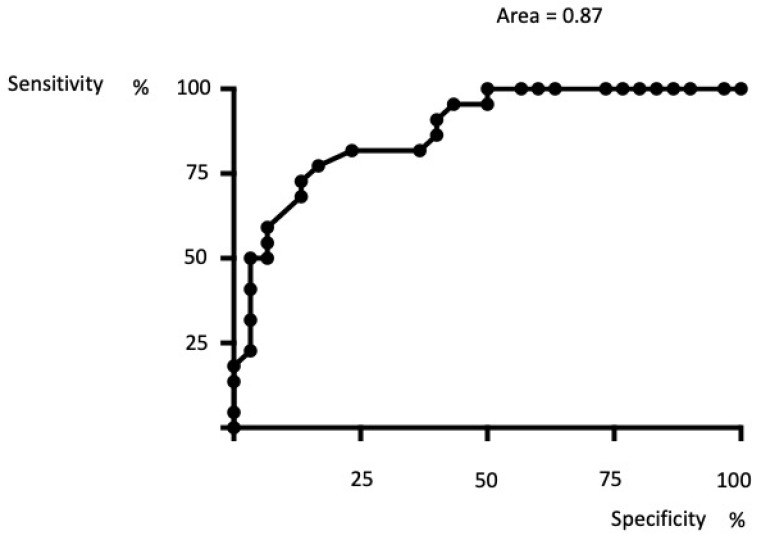
Receiver operating characteristic (ROC) curves and area under the receiver operating characteristic (AUROC) values. The ROC curve shows the trade-off between sensitivity (the true positive rate) and specificity (the false positive rate). Better ratio sensitivity/specificity was obtained for a cutoff value of the 72.5% inhibition in the novel ELISA. The area under the curve was well over 50% (0.87); therefore, the test can be considered to have good precision.

**Table 1 biomedicines-10-01849-t001:** Demographics, risk factors, angiographic data, and positive non-invasive diagnostic Group 1: obstructive CAD (FFR < 0.8); Group 2: non-obstructive CAD (FFR > 0.8).

	Group 1 N = 31	Group 2 N = 23	*p*
Female	10 (32.2%)	7 (30.4%)	NS
Age, year (mean ± SD)	67 ± 5.7	70 ± 7.6	NS
BMI kg/m^2^	29 ± 7.1	30 ± 5.9	NS
Cardiovascular risk factor			
Dyslipidemia	20 (64.5%)	11 (47.8%)	*p* < 0.05
Smoker Never and former Current	13 (41.9%) 18 (58.1%)	12 (52.2%) 11 (47.8%)	*p* < 0.05 *p* < 0.05
Diabetes	12 (38.7%)	4 (17.4%)	*p* < 0.05
None	19 (61.3%)	19 (82.6%)	*p* < 0.05
T1DM	2 (6.4%)	1 (4.7%)	NS
T2DM	10 (38.7%)	3 (13%)	*p* < 0.05
Family history of premature CAD	8 (25.8%)	6 (28.4%)	NS
HTA	17 (54.8%)	13 (56.5%)	NS
Treatment			
Calcium channel blockers	4 (12.9%)	3 (13%)	NS
Angiotensin converting enzyme inhibitor	12 (38.7%)	9 (39.1%)	NS
Angiotensin receptor blocker	8 (25.8%)	6 (26%)	NS
Beta-blockers	2 (6.4%)	1 (4.3%)	NS
Statins	19 (61.2%)	9 (39.1%)	*p* < 0.05
Metformin	10 (32.2%)	3 (13%)	*p* < 0.05
DPPIV inhibitors	9 (29%)	4 (17.3%)	*p* < 0.05
Insulin therapy	2 (6.4%)	1 (4.3%)	NS
Angiographic findings: Number of diseased vessels			
0		23	
1	18 (58.1)	0	
2	7 (22.6%)	0	
3	6 (19.3%)	0	
Culprit vessel			
Left main disease	1 (3.2%)	0	
Left anterior descending artery	20 (64.5%)	0	
Circonflex coronary artery	15 (48.38%)	0	
Right coronary artery	14 (45.1%)	0	
Abnormal noninvasive diagnostic testing			
Abnormal CTA	15 (48.3%)	11 (47.8%)	NS
Positive Stress echocardiography	7 (22.5%)	5 (21.7%)	NS
Positive Myocardial perfusion scintigraphy	10 (32.2%)	7 (30.4%)	NS

## Data Availability

The raw data supporting the conclusions of this manuscript can be made available by the authors without undue reservation.

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
