# Peer review of "Plasma A2AR Measurement Can Help Physicians Identify Patients Suspected of Coronary Chronic Syndrome: A Pilot Study"

_biomedicines, 2022, doi:10.3390/biomedicines10081849_

Round 1
Reviewer 1 Report
I have no further comments.
Author Response
Answer: Thank You
Reviewer 2 Report
The manuscript titled as "Plasma A2AR Measurement Can Help Physicians 2 Identify Patients Suspected of Coronary Chronic 3 Syndrome: A Pilot Study" used homemade inhibition ELISA to measure A2AR in plasma to evaluate patients with stable CCS.
In the previous study "Production of an agonist-like monoclonal antibody to the human A2A receptor of adenosine for clinical use." shows the innovative usage of A2AR antibody. I assumed that the aim of this study is to further prove the clinical value of A2AR inhibition ELISA. However, only around 30 samples were collected in each group in this study.
The results can be accepted, only if the samples in each group increase into 50+.
Author Response
The manuscript titled as "Plasma A2AR Measurement Can Help Physicians 2 Identify Patients Suspected of Coronary Chronic 3 Syndrome: A Pilot Study" used homemade inhibition ELISA to measure A2AR in plasma to evaluate patients with stable CCS.
In the previous study "Production of an agonist-like monoclonal antibody to the human A2A receptor of adenosine for clinical use." shows the innovative usage of A2AR antibody. I assumed that the aim of this study is to further prove the clinical value of A2AR inhibition ELISA. However, only around 30 samples were collected in each group in this study.
The results can be accepted, only if the samples in each group increase into 50+
Answer: As specified previously this is a pilot study. Furthermore the statistical analysis show significant Differences in soluble A2a concentration in plasma between groups.
Reviewer 3 Report
There is not a formal reply to my previous report (or it is not accessible to me), however authors changed the text accordingly. I have few other comments:
As already underlined in my previous report I would change “non-contributive electrocardiography and troponin” simply with “negative electrocardiography and troponin”.
Abstract continues to be confused in presentation as underlined in previous report.
In flow chart “figure 2” which is the difference between “met non inclusion criteria” and “met exclusion criteria”.
Author Response
here is not a formal reply to my previous report (or it is not accessible to me), however authors changed the text accordingly. I have few other comments:
Answer: Sorry
As already underlined in my previous report I would change “non-contributive electrocardiography and troponin” simply with “negative electrocardiography and troponin”.
Answer: this was done
Abstract continues to be confused in presentation as underlined in previous report.
In flow chart “figure 2” which is the difference between “met non inclusion criteria” and “met exclusion criteria”.
Answer: met non inclusion criteria means that patient do not meet the conditions to be included in the study. For example presence of an associated chronic pathology. Met exclusion criteria means that patient was first included and finally excluded for example because of a rise in troponin.